

# Combined effects of grain size, flow volume and channel width on geophysical flow mobility: 3-D discrete element modeling of dry and dense flows of angular rock fragments

Bruno Cagnoli[1], and Antonio Piersanti[2]

[1]Istituto Nazionale di Geofisica e Vulcanologia, Via Donato Creti 12, 40128 Bologna, Italy.

[2]Istituto Nazionale di Geofisica e Vulcanologia, Via di Vigna Murata 605, 00143 Rome, Italy.

*Correspondence to:* Bruno Cagnoli (bruno.cagnoli@ingv.it)

**Abstract.** We have carried out 3-D numerical simulations by using a discrete element method (DEM) to study the mobility of dry granular flows of angular rock fragments. These simulations are relevant for geophysical flows such as rock avalanches. The model is validated by previous laboratory experiments. We show that: 1) the finer the grain size, the larger the mobility of the center of mass of granular flows, 2) the smaller the flow volume, the larger the mobility of the center of mass of granular flows and 3) the wider the channel, the larger the mobility of the center of mass of granular flows. The grain size effect is due to the fact that finer grain size flows dissipate intrinsically less energy. This volume effect is the opposite of that experienced by the flow fronts. Six different channel cross sections are tested. We introduce here a new scaling parameter $\chi$ that has the product of grain size and the cubic root of flow volume at the numerator and the product of channel width and flow length at the denominator. The linear correlation between the reciprocal of mobility and parameter $\chi$ is statistically highly significant. Parameter $\chi$ implies that the mobility of the center of mass of granular flows is an increasing function of the ratio of the number of fragments per unit of flow mass to the total number of fragments in the flow. These are two characteristic numbers of particles whose effect on mobility is scale invariant.

*Keywords*: Discrete Element Modeling; Mobility of Geophysical Flow; Grain Size; Flow Volume; Channel Width

## 1 Introduction

The prediction of mobility of geophysical flows is one of the most important research goals in the earth sciences. The geophysical flows we are interested in here include rock avalanches (Hungr et al., 2014) and the underflow of pyroclastic flows such as block-and-ash flows (Cas and Wright, 1988). These flows of angular rock fragments are dense and dry and they can be considered at the top of the list of the most hazardous natural phenomena. Rock avalanches are dry because their extensive fragmentation during motion generates new pore spaces that cannot be filled by water during their relatively short travel times (Hungr et al., 2014). Block-and-ash flows are dry because they are due to the collapse of volcanic domes (Nairn



and Self, 1978; Saucedo et al., 2002) or the explosive fragmentation of volcanic plugs (Hall et al., 2015) where domes and plugs are too hot for liquid water to be present.

Grain size, flow volume and channel width are among the main variables that affect the mobility of geophysical flows. This is due to the fact that these quantities can vary significantly in nature and this causes significantly different travel
distances. In the present paper, we illustrate the combined effect of grain size, flow volume and channel width on granular flow mobility. This result is built on our previous laboratory experiments (Cagnoli and Romano, 2010, 2012a) and numerical simulations (Cagnoli and Piersanti, 2015) where we focused on the quantities that enter the numerator of a scaling parameter that the apparent coefficient of friction $\mu_A$ (i.e., the reciprocal of mobility) is proportional to. Here we study the effect on flow mobility of the channel width, which enters the denominator of this scaling parameter.
Our research shows that the finer the grain size (all the other features being the same), the more mobile the center of mass of the granular flows. We expect for example that, in nature, the fragmentation during motion of rock clasts with different hardness can generate flows with different grain size. There is field evidence showing that rock avalanches of harder rocks tend to be less mobile than those of weaker rocks (Zhang and Yin, 2013). Importantly, we demonstrate that finer grains size flows are inherently less energetically dissipative than coarser grain size flows irrespective of the presence of an interstitial
fluid (either gaseous or liquid).

Our research shows also that the larger the flow volume (all the other features being the same), the less mobile the center of mass of the granular flows. This phenomenon (which was observed also by Okura et al., 2000) does not contradict the conclusions obtained by Scheidegger (1973). In the paper by Scheidegger (1973), his apparent coefficient of friction decreases as flow volume increases because his coefficient is computed taking into account the frontal end of the deposits
which is more distal as flow volume increases. This, which is true in our system as well (Cagnoli and Romano, 2012a), is due to the fact that the larger the flow volume, the larger the longitudinal spreading of the deposit (Davies, 1982; D'Agostino et al., 2010). Similarly, the planimetric area inundated by a flow is proportional to a power of the flow volume (Griswold and Iverson, 2008).

Here we illustrate the results of 3-D numerical simulations carried out by means of a discrete element method (DEM) that
we have shown to predict correctly the relative mobility of laboratory granular flows with different features (Cagnoli and Piersanti, 2015). It is because of this previous experimental validation of our computer modeling that we rely on its predictive power to replace labour-intensive and time-consuming laboratory experiments. This is particularly useful in this paper where we compare the mobility of flows in six different channels with different cross sections whose construction and use in laboratory experiments would be truly demanding. Discrete element modeling considers particle-particle and particle-
boundary interactions and, for this reason, it is able to estimate correctly the relative energy dissipation of the flows without prior parameter tuning. Studies of granular flows by means of discrete element modeling include Valentino et al. (2008), Banton et al. (2009), Girolami et al. (2012), Yohannes et al. (2012), Mollon et al. (2012), Cagnoli and Piersanti (2015) and Mead and Cleary (2015).



Our numerical simulations show that the narrower the channel (all the other features being the same), the smaller the mobility of the center of mass of the granular flows. Therefore the channel width (together with the length of the flow or that of the deposit) enters the denominator of a new scaling parameter $\chi$ that gives rise to a highly significant linear correlation with the reciprocal of mobility $\mu_A$. Parameter $\chi$ implies that the mobility of the center of mass of granular flows is an

5 increasing function of the ratio of the number of particles per unit of flow mass to the total number of particles in the flow.

Although there are many more variables (besides grain size, flow volume and channel width) that can affect the mobility of geophysical flows in nature (such as the initial speed, the subsurface friction, the intergranular fluid density etc.), the fact that their values are constant or equal to zero in our analysis is not a shortcoming. In fact, to be useful, a model has to single out a single phenomenon. The introduction of too much complexity can hinder the comprehension of the effect of a variable

in a system where different phenomena interfere. It is also not the actual size of the examined granular flows (either in a laboratory experiment or numerical simulation) that matters. Even if our system was ten times larger (assuming to be able to solve the consequent practical complications in the laboratory and numerical problems in the computer simulations, such as prohibitively large computer processing times), it would still be much smaller than the corresponding natural system. The attention of the reader should rather focus on whether the phenomenon that the model is focusing on is scale invariant. The

grain size effect, the volume effect and the channel width effect that are illustrated here are scale invariant because they are a function of only characteristic numbers of particles which, since they are dimensionless, determine the distinctive character of any granular flow at any scale either in the laboratory or in nature.

## 2 Method

### 2.1. Channels and Granular Material in the 3-D Numerical Simulations

In our 3-D numerical simulations, we vary the width $w$ of the concave-upward channel we have used in our previous laboratory experiments (Cagnoli and Romano, 2012a) and numerical simulations (Cagnoli and Piersanti, 2015). These virtual channels have been generated by means of a CAD software (Rhinoceros). Both here and in our previous works, the channels consist of a straight upper ramp and a curved chute (Figs. 1 and 2). The longitudinal profile of the curved chutes is computed by using a hyperbolic sine equation:

$z = 0.3 - 0.085 \operatorname{arcsinh}(11.765\,x) ,$ (1)

where the variables are in meters. This equation represents a slightly modified version of the profile of Mayon volcano in the Philippines (Becker, 1905). The horizontal length of the curved chutes is 1.4 m. The granular material is placed behind a sliding gate located 22.3 cm above x=0, where this distance is measured along the upper ramp (Figs. 1 and 2). The gate is removed in a direction perpendicular to the surface of the upper ramp.

The upper ramps and the chutes have the same trapezoidal cross section (Fig. 1) that corresponds, in nature, to a V-shaped topographic incision with sediment infilling in the centre (see, for example, the natural channel cross sections in





Zhang and Yin, 2013). The width $w$ we use in the calculations is the smallest distance between the inclined sidewalls of the channels (Fig. 1). This distance has been preferred because the width at the base of a channel that is trapezoidal in cross section concerns the flows more than that at the top that is larger than the flow width. Here we adopt $w$ values equal to 6 mm (the same value used in our previous publications), 10 mm, 16 mm and 26 mm (Fig. 3). Most of the simulations (sixteen) have been carried out in channels with the same lateral side inclination $\theta$ (27º), but two ancillary simulations with smaller and larger $\theta$ values (19º and 41°) have also been carried out (Fig. 3).

The numerical flows are dry and consist of particles with three different shapes. We use particles with a cubic shape, half a cubic shape and a quarter of a cubic shape (Fig. 4). These polyhedrons represent an equant, an oblate and a prolate particle, respectively. Non-spherical particles are preferred because when interacting among themselves and with the boundary surfaces, their energy dissipation mechanism during collision and attrition is comparable with that of natural fragments since they are both angular (Cagnoli and Romano, 2012a; Mead and Cleary, 2015). The proportion of each particle shape is always the same in all flows irrespective of grain size, flow mass or channel features: the equant particles are 38%, the oblate particles 22% and the prolate particles 40% of the flow mass. This generates more realistic flows than those with particles of only one shape, because natural geophysical flows contain particles with different shapes. We have shown that flows with different proportions of particle shapes dissipate different amounts of energy per unit of travel distance (Cagnoli and Piersanti, 2015). For this reason, it is important that the comparison of mobility is carried out among flows with the same proportions of particle shapes. Concerning grain size, we adopt geometrically similar polyhedrons whose longer edges can be 1, 1.5, or 2 mm in length. Only one grain size is used in each granular flow.

Table 1 illustrates the combinations of the values of the channel width $w$, the lateral side inclination $\theta$, the grain size $\delta$ and the total granular flow mass adopted in the numerical simulations. In the channel with $w = 6$ mm and $\theta = 27º$, we use total granular flow masses equal to 8.9, 13 and 26 g. This total mass increases in wider channels with $\theta = 27º$: 29 g when $w = 10$ mm, 33 g and 68 g when $w = 16$ mm, 39 g and 93 g when $w = 26$ mm. The flow mobility of simulations with sidewall inclinations $\theta$ equal to 19º and 41° (where $w = 6$ mm and $\delta = 1$ mm) is meant to be compared with the mobility obtained with $\theta = 27º$ and the same channel width and the same grain size. In the simulations with $\theta$ equal to 19º and 41°, the total granular flow mass is 36 and 17 g, respectively. We thus test also a relatively large range of total granular mass values.

Table 2 shows the values of the physical properties of particles, channels and gates adopted in the numerical simulations. The properties of the particles are those of an igneous rock, the properties of the channels are those of clay and the properties of the gates are those of aluminum. Table 3 shows the values of the properties that govern particle-particle, particle-channel and particle-gate interactions. These values indicate that we are simulating flows of rock fragments that travel on a subsurface made of soil (Peng, 2000). The angle of internal friction of the granular material is not explicitly mentioned by the model, but, this important property is determined by the shapes of the particles. Our numerical simulations pertain to angular fragments. The surface of the gates has a very small friction value to avoid disturbing the granular material when the gates are removed. The roughness of the chute and ramp surfaces is identical and it is everywhere significantly smaller than the smallest grain size we use.



In our software (EDEM), we need to locate the particles at time zero. For this purpose, we generate, behind the gate, 3-D abstract spaces that are filled (before the gate is removed) with particles in random position and without interpenetration. These abstract spaces do not represent any real material object. We use abstract spaces whose volumes are proportional to the granular masses so that the same compaction and density of the granular material behind the gate before release is

obtained. We have shown that the larger the initial compaction of the granular material behind the gate before release, the larger the mobility of the center of mass of the granular flow (Cagnoli and Piersanti, 2015). Therefore it is important that the comparison of mobility is carried out among granular masses that before release have the same initial compaction. Within our abstract spaces, the particles are in contact with one another and the granular masses have a bulk average density equal to 721±14 kg/m$^3$. The density of the particle material is 2700 kg/m$^3$ (Table 2). In the numerical simulations we use granular

masses that are relatively small so that the computer processing times are manageable. The flows with the largest number of particles (~34000) are those with a total mass equal to 39 g and a grain size equal to 1 mm. The flows with the smallest number of particles (~2800) are those with a total mass equal to 26 g and a grain size equal to 2 mm.

### 2.2. Contact Model of the Numerical Simulations

Our 3-D discrete element modeling has been carried out by using the software EDEM developed by DEM Solutions

(www.dem-solutions.com). The approach that EDEM adopts when dealing with particles is twofold. On one hand, it adopts the mass, volume and moment of inertia of the polyhedrons we have chosen. On the other hand, it uses clusters of spheres inscribed within the polyhedrons (Fig. 4) to estimate impact forces during particle collisions. These forces are a function of sphere overlaps. Experience has shown that clusters of spheres are an effective method to model complex particle shapes with a good degree of approximation (DEM Solutions, 2014). This method allows good computing performance because the

contact detection algorithm for clusters of spheres is more efficient than that for polyhedrons. For particle collisions, the model computes normal and tangential forces, their damping components and the tangential and rolling friction forces (DEM Solutions, 2014).

The normal force (Hertz, 1882) is

$$F_n = \frac{4}{3} E^* \sqrt{R^*} \lambda_n^{3/2} , \qquad (2)$$

where $\lambda_n$ is the normal overlap, $E^*$ is the equivalent Young's modulus and $R^*$ is the equivalent radius that are defined as follows:

$$\frac{1}{E^*} = \frac{(1-\nu_i^2)}{E_i} + \frac{(1-\nu_j^2)}{E_j} \qquad (3)$$

and

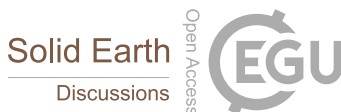



$$\frac{1}{R^*} = \frac{1}{R_i} + \frac{1}{R_j} , \tag{4}$$

respectively. Here, $E$ and $\nu$ are the Young's moduli and the Poisson's ratios, respectively, of the interacting elements i and j (polyhedrons, channel or gate). $R_i$ and $R_j$ are the radii of the interacting spheres of the interacting polyhedrons i and j. When one of the two interacting elements is not a particle, the equivalent radius is equal to the radius of the interacting sphere of

the polyhedron.

The tangential force (Mindlin, 1949; Mindlin and Deresiewicz, 1953) is

$$F_t = -S_t \lambda_t , \tag{5}$$

where $\lambda_t$ is the tangential overlap and $S_t$ is the tangential stiffness which is a function of the equivalent shear modulus $G^*$. The stiffness is

$$S_t = 8G^* \sqrt{R^* \lambda_n} \tag{6}$$

and the equivalent shear modulus is

$$\frac{1}{G^*} = \frac{(2 - \nu_i)}{G_i} + \frac{(2 - \nu_j)}{G_j} , \tag{7}$$

where $G_i$ and $G_j$ are the shear moduli of the interacting elements. The tangential force is limited by Coulomb's friction which is equal to

$$\mu_s F_n , \tag{8}$$

where $\mu_s$ is the coefficient of static friction (Cundall and Strack, 1979).

The normal and tangential damping components (Tsuji et al., 1992) are

$$F_n^d = -2 \sqrt{\frac{5}{6}} \varepsilon \sqrt{S_n m^*} u_n^{rel} \tag{9}$$

and

$$F_t^d = -2 \sqrt{\frac{5}{6}} \varepsilon \sqrt{S_t m^*} u_t^{rel} , \tag{10}$$

respectively. In Eqs. (9) and (10),

$$\varepsilon = \frac{\ln e}{\sqrt{\ln^2 e + \pi^2}} , \tag{11}$$





$$m^* = \left( \frac{1}{m_i} + \frac{1}{m_j} \right)^{-1} \quad (12)$$

and

$$S_n = 2E^* \sqrt{R^* \lambda_n} \,, \quad (13)$$

where $m^*$ is the equivalent mass, $m_i$ and $m_j$ are the masses of the interacting elements (polyhedrons, channels or gates), $e$ is the coefficient of restitution, $S_n$ is the normal stiffness and the $u^{rel}$ values are the normal (subscript n) and tangential (subscript t) components of the relative velocity.

The rolling friction (Sakaguchi et al., 1993) is accounted for by applying a torque

$$\tau_i = -\mu_r F_n d_i \omega_i \quad (14)$$

to the contacting surfaces. Here $\mu_r$ is the coefficient of rolling friction, $d_i$ is the distance from the centre of mass of the polyhedron to the contact point (where the contact point is defined in the middle of the overlap) and $\omega_i$ is the unit angular velocity vector of the particle at the contact point. This torque is calculated independently for each polyhedron.

## 3 Scaling

We measure the reciprocal of mobility of a flow by using the apparent coefficient of friction

$$\mu_A = \frac{h}{l}, \quad (15)$$

where $h$ is the vertical drop of the centre of mass of the granular material and $l$ is its horizontal distance of travel. Distances $h$ and $l$ are measured from the position of the centre of mass of the granular samples at rest behind the gate before release to the position of the centre of mass of the final deposits (Fig. 1). In an energy dissipation (i.e., mobility) analysis, the knowledge of the whereabouts of the center of mass is a necessity because the center of mass is the only point that is a proxy of the entire flow since it moves as though the total mass of all the particles were concentrated there and all the external forces were applied there. Our parameter, thus, differs from the so-called Heim coefficient that considers the vertical and horizontal distances between the highest point of failure and the most distal position of the front of the deposit (Scheidegger, 1973).

In both the laboratory experiments (Cagnoli and Romano, 2012a) and the numerical simulations (here and in Cagnoli and Piersanti, 2015), the deposited granular material consists of two portions: a more proximal heap that is much more elongated than thick (the deposit of the flow proper) and a more distal distribution of individual fragments. The distal distribution is



formed by fragments, which, bouncing within the chute, traveled individually without interacting and are not part of the flow proper. Flows and distal distributions have different movement and depositional mechanisms and they must be considered separately. Here we study the flow proper. Therefore, the value of $\mu_A$ is computed for the final deposit of the flow proper, which consists of all the particles on the chute that are in contact with one another. The particles of the distal distribution are

those that are not in contact with one another.

We are interested here in a functional relation between the apparent coefficient of friction $\mu_A$, the grain size $\delta$, the flow volume $V$, the channel width $w$ and the length of the flow $L$:

$$f(\mu_A, \delta, V, w, L) = 0 . \tag{16}$$

The length of the flow $L$ is considered the pertinent length scale for the flow as a whole (Iverson et al., 2010). We introduce

$L$ here to include in the analysis the extent of the entire spread of the flow particles along the longitudinal direction of the channel. The width $w$ gives the same information along the transversal direction. The combined presence of $L$ and $w$ is instrumental to provide information on the number density of the particles within the flows. In general, granular flow mobility is affected by many more quantities whose values are, however, held deliberately constant in our system as discussed in Section 5.4.

Equation (16) has 5 quantities with one fundamental physical dimension (length). For this reason, according to the Buckingham Pi Theorem (e.g., Dym, 2004), it is equivalent to a functional relation containing four dimensionless parameters. Since the grain size $\delta$ is a key quantity ($\delta$ is the pertinent length scale for the grain-scale mechanics that generates the stresses (Iverson et al., 2010)), we use it to scale the other variables so that Eq. (16) is replaced by the following functional relation $F$:

$$\mu_A = F\left(\frac{V^{1/3}}{\delta}, \frac{L}{\delta}, \frac{w}{\delta}\right). \tag{17}$$

Our numerical simulations (section 4) show that the use of these dimensionless parameters results in a good collapse of the data along a single fitting straight line:

$$\mu_A = a\chi + b \tag{18}$$

where

$$\chi = \frac{V^{1/3}\delta}{Lw} . \tag{19}$$

Parameter $\chi$ is a ratio of the three independent dimensionless parameters of Eq. (17) and it can be expressed as the ratio of quantities $\Gamma$ and $\gamma$:





$$\chi = \frac{V^{1/3}\delta}{Lw} = \frac{V^{1/3}}{\delta}\frac{\delta}{L}\frac{\delta}{w} = \frac{V^{1/3}}{\delta}\frac{\delta^2}{Lw} = \frac{\Gamma}{\gamma} \qquad (20)$$

where

$$\Gamma = \frac{V^{1/3}}{\delta} \qquad (21)$$

and

5 $$\gamma = \frac{Lw}{\delta^2}. \qquad (22)$$

The dimensionless quantity $\Gamma$ is an increasing function of the total number of particles in the flow because it corresponds to $V/\delta^3$. The dimensionless quantity $\gamma$ is an increasing function of the number of particles per unit of flow volume (i.e., flow mass). This is so because, $\gamma$ is proportional to the number of particles whose sum of cross-sectional area covers completely a characteristic surface area of the flow equal to $Lw$ (the central cross-sectional area of a particle is proportional to $\delta^2$ ). Here

quantity $V$ is obtained by summing the individual volumes of all the particles that form the deposit of the flow proper so that, since the rock density is a constant, this $V$ value is proportional to the flow mass.

**4 Results**

The flows can be examined in cross-section (Figs. 5 and 6) or from the top (Figs. 7 and 8) because the numerical simulations are three-dimensional. As soon as the gate is removed, the granular mass accelerates down the slope, reaches a maximum

speed and then it decelerates and stops (Fig. 5). Figure 6 shows that the slope-parallel component of particle velocity within the flows decreases toward the subsurface along its normal as expected in flows that travel in contact with a boundary surface (Iverson, 2003; Cagnoli and Romano, 2013). Particles that are not part of the flow proper because they travel individually and form, when at rest, the distal distribution are clearly discernible in the numerical simulations at the front and at the back of the flow proper (Figs. 7 and 8).

The reciprocal of mobility $\mu_A$ is plotted versus parameter $\chi$ in Figs. 9 and 10. The values of $\mu_A$ in these figures are relatively large because they refer to the final position of the center of mass and not to the final position of the front of the deposit (the flow front is more mobile than the center of mass). We have also adopted a relatively large basal friction value (Table 3). The collapse along a single straight line of all the data points of the simulations with $\theta = 27^\circ$ confirms that, in Figs. 9 and 10, only the variables considered in Eq. (19) have values that vary and, consequently, determine the observed different

mobility of the centre of mass of the different flows.





In the figures of this paper, we have added to each $\mu_A$ value of the flow proper an uncertainty bar whose extremities are computed as follows. The upper extremity is the $h/l$ value of the portion of deposit whose distal end is 3 cm in a more proximal position than the distal end of our best estimate of the deposit of the flow proper. The lower extremity is the $h/l$ value of all the particles on the chute (deposit of the flow proper plus distal distribution). Even if there is some uncertainty

when locating the position of the distal end of the deposit of the flow proper, it is clear that the 3 cm shorter deposit and all the particles on the chute provide $h/l$ values that can be significantly different from the true $\mu_A$ value of the flow proper. The likelihood that the true $\mu_A$ value of the flow proper is located somewhere along the uncertainty bar decreases quickly to zero at a relatively short distance from our best estimate.

In Fig. 9, parameter $\chi$ is computed by setting $L$ equal to the length of the deposits. In this figure, Eq. (18) has a

coefficient $a$ equal to 0.95 and a coefficient $b$ equal to 0.81. The linear correlation coefficient $r$ (Taylor, 1997) of the fitting straight line in this figure is equal to 0.93. Since the probability that 16 values of two uncorrelated variables generate an $r$ value equal to 0.93 is much less than 0.05% (Taylor, 1997), the linear correlation between $\mu_A$ and $\chi$ in Fig. 9 is highly significant.

In Fig. 10, parameter $\chi$ is computed by setting $L$ equal to the length of the flows (measured 0.58 sec after the gate

removal in all simulations). In this figure, Eq. (18) has a coefficient $a$ equal to 2.05 and a coefficient $b$ equal to 0.81. The linear correlation coefficient $r$ (Taylor, 1997) of this fitting straight line is equal to 0.91. Since the probability that 16 values of two uncorrelated variables generate an $r$ value equal to 0.91 is less than 0.05% (Taylor, 1997), also the linear correlation between $\mu_A$ and $\chi$ in Fig. 10 is highly significant.

In Figs. 9 and 10, the straight line fits only the simulation results in channels with different width $w$ but the same sidewall

inclination $\theta$ (in this case equal to 27º). Simulations results in channels with different sidewall inclinations (here 19º and 41º) plot along different straight lines (we expect one straight line for each sidewall inclination) since parameter $\chi$ does not consider $\theta$. Figures 9 and 10 show that the larger the value of the sidewall inclination $\theta$, the larger the value of $\mu_A$.

## 5. Discussion

### 5.1. Grain Size Effect on Flow Mobility

The results of the numerical simulations illustrated in this paper (Figs. 9 and 10) show that the finer the grain size (all the other features being the same), the more mobile the centre of mass of the granular flows (Cagnoli and Romano, 2010, 2012a). This is due to the fact that, the finer the grain size, the less agitated the particles per unit of flow mass so that, finer grain size flows dissipate less energy per unit of travel distance (Cagnoli and Romano, 2010). Particles are less agitated, because, in finer grain size flows (all the other features being equal, including the total flow mass), there is a larger number

of clasts per unit of flow mass and, for this reason, the agitation due to the interaction with the rough subsurface and the



rough sidewalls penetrates relatively less inside the flows (Fig. 11). It is for this reason that the reciprocal of mobility $\mu_A$ is inversely proportional to $\gamma$ (Eq. (18)), where $\gamma$ is an increasing function of the number of particles per unit of flow mass.

In the laboratory, the increase of particle agitation as grain size increases (all the other features being the same) is noticeable visually in the high speed video camera images of the granular flows and it has been assessed quantitatively: 1) by measuring particle agitation as the normalised average squared deviation from the mean of the particle transversal speeds and 2) by measuring the fluctuations of granular pressure at the base of the flows (Cagnoli and Romano, 2010; 2012b). Also speed and energy calculations in numerical simulations confirm that flows with coarser fragments have more agitated particles and that they are energetically more dissipative than flows of finer particles with all the other features being the same (Cagnoli and Piersanti, 2015). Particle agitation affects energy dissipation through particle-particle and particle-boundary interactions (friction and collisions) and through diversion of energy into directions (the transversal one for example) that are different from the downslope direction.

Figures 9 and 10 document the change in mobility of the centre of mass as the granular flows experience the internal structural change caused by the change in grain size. In particular, the finer the grain size (all the other features being the same), the smaller the agitation of the particles per unit of flow mass and, thus, the thicker the plug, where the plug is the portion of the flow with a much smaller (but not necessarily zero) particle agitation (Cagnoli and Romano, 2012a, 2012b). In rapid granular flows, when the plug is present, most particle agitation is located in a relatively thin layer in contact with the subsurface and the sidewalls (Cagnoli and Romano, 2013). Thus, Eq. (18) shows that rapid granular flows with a plug have a relatively more mobile centre of mass.

Numerical simulations (Mollon et al., 2015) have shown that also on laterally unconfined planar slopes ending on a depositional horizontal plane with no lateral constraints, an increase of particle size tends to decrease the mobility of the center of mass when a collisional regime is allowed to occur. In debris flows, the presence of intergranular clay and water can alter this trend because of the activation of a different rheology as the fraction of the coarse grains decreases. De Haas et al. (2015) found experimentally that, in debris flows, as the content of coarse grains (gravel in their case) decreases, the remaining slurry is not able to build lateral levees so that, below a critical gravel content, the lateral spreading on unconfined slopes increases and, consequently, the longitudinal runout decreases.

**5.2 Flow Volume Effect on Flow Mobility**

Figures 9 and 10 show also that there is a decrease of mobility of the centre of mass as flow volume increases with all the other features being the same (Cagnoli and Romano, 2012a; Cagnoli and Piersanti, 2015). This effect is the result of the fact that a deposit accretes backward during its formation on a change of slope. In this case, the larger the quantity of granular material that accumulates at the back (i.e., the larger the flow volume), the longer the backward shift of the center of mass (Cagnoli and Romano, 2012a). It is for this reason that the reciprocal of mobility $\mu_A$ is directly proportional to $\Gamma$ (Eq. (18)), where $\Gamma$ is an increasing function of the total number of particles in the flow. The backward accretion is due to the fact that the front of the flow reaches the less steep part of the curved slope and stop before the rear part, preventing the rear part, and




the centre of mass, from travelling further downhill as revealed by high-speed video camera images in the laboratory experiments and as clearly visible in the numerical simulations.

The volume effect we observe occurs whenever deposition takes place on a slope change, either gradual (here and in Cagnoli and Romano, 2012a) or abrupt (Okura et al., 2000). Okura et al. (2000) obtained this effect on a laterally unconfined

planar slope ending on a horizontal depositional plane with no lateral constraints. With such sharp breaks in slope, the larger the change of direction, the larger the energy dissipation of the flow (Manzella and Labiouse, 2013). In nature, the flank of a volcano provides an instance of a gradual change of the slope inclination angle (besides the flanks of Mayon Volcano, other examples are those of Mt. Ngauruhoe (Lube et al., 2007) and Volcán de Colima (Saucedo et al., 2002)), whereas the intersection of a mountain side with an alluvial plain is an example of an abrupt change of the slope inclination angle. When

assessing the volume effect in laboratory experiments and numerical simulations, the measure of mobility of the flows must take into account that within the same container at the top of the slope before release, larger volumes have an unavoidable higher elevation of their center of mass for a geometric reason. In parameter $\mu_A$ (Eq. (15)), flow mobility is indeed normalised by the different initial elevations $h$ of the center of mass of the different granular volumes.

**5.3 Channel Width Effect on Flow Mobility**

In our previous publications (Cagnoli and Romano, 2012a; Cagnoli and Piersanti, 2015), we introduced a scaling parameter $\beta = \delta V^{1/3}/w^2$ , which has the same numerator of $\chi$, but a different denominator. These previous publications focused exclusively on the effects of grain size $\delta$ and flow volume $V$ (i.e., the numerator of $\beta$). The denominator of $\beta$ was conjectural since the channel width $w$ did not vary. This conjecture was however based on the observation that, with a change in slope, the front of a granular flow stops before the rear, so that the deposit propagates backward and, the narrower

the channel, the longer the deposit and, thus, the longer the backward migration of the centres of mass during deposition. In other words, although the channel width is expected to be at the denominator because the smaller its value, the smaller the mobility of the center of mass, a specific set of investigations where the value of $w$ varies systematically is necessary to figure out its exponent.

The plot of $\mu_A$ versus $\beta$ confirms that the exponent of quantity $w$ is different from 2. In Fig. 12, simulations with the same

channel width plot along the same straight line (thus confirming the same grain size and flow volume effects of Figs. 9 and 10), but the straight lines for different channel widths have different gradients. Therefore, in parameter $\beta$, an exponent of $w$ equal to 2 falls short of accommodating satisfactorily the channel width effect. For example, if we drew a fitting straight line of all the simulations with $\theta = 27°$ in Fig. 12, it would have a linear correlation coefficient $r$ (Taylor, 1997) equal to only 0.75.

This paper shows that the exponent of $w$ is equal to 1 so that another quantity has to be introduced at the denominator of $\chi$ for this parameter to be dimensionless. This quantity $L$ is here either the length of the deposits (Fig. 9) or the length of the flows (Fig. 10). Both length values can be used because, when comparing model and nature, one is expected to scale as the





other does. The length of the flow is however difficult to assess in the field (where pyroclastic flows and rock avalanches are hidden by clouds of fine particles and they are too dangerous to approach at close range) and in the laboratory or in the numerical simulations (where there are numerous isolated particles at the front and at the back of the flows so that it is difficult to decide where the transition between the different transport mechanisms of saltating particles and the dense flow

proper occurs). It is for this reason that in Fig. 10, where $\chi$ is computed by using the length of the flows, the linear correlation coefficient is smaller than that in Fig. 9 where $r$ is computed by using the length of the deposits.

It is important to realize that parameters $\chi$ and $\beta$ have the same meaning because they both imply that the mobility of the centre of mass of the granular flows is an increasing function of the ratio of the number of particles per unit of flow mass to the total number of fragments in the flow (Cagnoli and Romano, 2012a). Notably, the characteristic surface area $Lw$ in $\chi$ is

actually more appropriate than $w^2$ in $\beta$ to uphold that

$$\mu_A \propto \frac{\Gamma}{\gamma}, \qquad (23)$$

because $Lw$ considers the complete spread of all the particles along the entire flow length. Quantity $L$ cannot be set directly as an initial condition by the experimenter in the laboratory or the modeller in the numerical simulations, because it is determined by the other channel and flow features that govern the spread of the particles within the channel. In other words:

$$L = \frac{1}{\chi} \frac{V^{1/3}\delta}{w}, \qquad (24)$$

where $1/\chi$ is a proportionality constant. Quantity $L$ is however useful in Eq. (16) because the knowledge alone of $V$ and $w$ does not suffice to reckon the extent of the longitudinal spread of the particles within the channel since the thickness of the flow is missing.

**5.4 Other Quantities Affecting Flow Mobility**

In general, beside grain size, flow volume and channel width, the mobility of the centre of mass of a granular flow is a function of several other variables as well. These variables include: initial flow speed $s$ (which is equal to zero in all our flows), acceleration of gravity $g$, density $\rho_s$ of the particles, density $\rho_f$ of the intergranular fluid, dynamic viscosity $\eta$ of the intergranular fluid, angle of internal friction $\phi$ of the particles, coefficient of restitution $e$ of the rock material, average height $i$ (the roughness) of the subsurface and sidewalls asperities, shape and surface texture of the particles and intergranular fluid

pressure. However, in our calculations, these variables have either a constant value (such as $s$, $g$, $\rho_s$, $\phi$, $e$, $i$ and the shape and the surface texture of the grains) or they refer to an intergranular fluid ($\rho_f$, $\eta$ and its pressure) that is absent in the numerical simulations either as a liquid or as a gas.

Concerning the intergranular fluid, the numerical simulations described in this paper demonstrate that the grain size effect and the flow volume effect we observe in the laboratory (Cagnoli and Romano, 2012a) are not due to the presence of either an intergranular gas or an intergranular liquid. This is so because the numerical simulations where an intergranular fluid is absent do, nonetheless, predict the same relative flow mobility observed in the laboratory (here and Cagnoli and
Piersanti, 2015). Therefore, importantly, finer grain size flows dissipate intrinsically less energy than coarser grain size flows irrespective of the presence of an intergranular fluid (either gaseous or liquid).

## 6. Conclusions

Our 3-D numerical simulations of granular flows of angular rock fragments show that (all the other features being equal): 1) the finer the grain size, the larger the mobility of their center of mass; 2) the smaller the flow volume, the larger the mobility
of their center of mass; 3) the wider the channel, the larger the mobility of their center of mass. Our model granular flows are relevant for natural systems where the flows acquire a fully developed travel shape after the initial deformation of the collapsing landscape feature (such as a mountain side or a volcanic dome) and before the final deposition. This fully developed shape of travel consists of a finite contiguous body of rock fragments traveling in unison. The grain size, the flow volume and the channel width effects illustrated here are scale invariant because they are a function of only characteristic
numbers of particles that, since they are dimensionless, determine the distinctive character of any flow at any scale as suggested also by the generalisation provided by the numerical simulations. In particular, the mobility of the center of mass of the granular flows is an increasing function of the ratio of the number of particles per unit of flow mass to the total number of particles in the flow.

*Data availability.* The data upon which the conclusions of this paper are based are available from the corresponding author on request.

*Acknowledgments.*

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

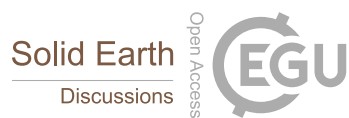

CAPTIONS

**Figure 1.** Longitudinal cross section of all the channels used in the numerical simulations. The inset illustrates the transversal cross section of both the straight ramps and the curved chutes where quantities $w$ (the channel width) and $\theta$ (the sidewall inclination) are shown. CM stands for centre of mass.

**Figure 2.** Example of a 3-D virtual channel generated by means of a CAD software (Rhinoceros) for the numerical simulations. The concave upward channel is shown in red and the gate in yellow.

**Figure 3.** The different values of the channel width $w$ and the sidewall inclination $\theta$ that are adopted in the numerical simulations.

**Figure 4.** Shapes of the particles that are used in the numerical simulations. The inscribed clusters of spheres are also shown.

**Figure 5.** Cross sections illustrating three moments of a flow history. The frame at the top shows the granular material at rest behind the gate before release. The frame in the middle shows the fully developed shape of travel of the granular flow. The lower frame shows the final deposit. In this example the granular mass is 93 g, the grain size is 2 mm, $w$ is 26 mm and $\theta$ is 27º. The values in seconds are the time measured after the gate removal.

**Figure 6.** Side view of a granular flow. Particle speed decreases toward the subsurface along its normal. In this example the granular mass is 26 g, the grain size is 1 mm, $w$ is 6 mm and $\theta$ is 27º. The time in seconds is measured after the gate removal.

**Figure 7.** Fully developed shape of travel of a granular flow seen from above 0.51 s after the gate removal. In this example, the granular mass is 26 g, the grain size is 1 mm, $w$ is 6 mm and $\theta$ is 27º.

**Figure 8.** Flow front seen from above 0.58 s after the gate removal. The saltating particles of the distal distribution in front of the dense granular flow are clearly identifiable. In this example the granular mass is 26 g, the grain size is 1 mm, $w$ is 6 mm and $\theta$ is 27º.

**Figure 9.** Plot of the reciprocal of mobility $\mu_A$ versus parameter $\chi$ where $L$ is set equal to the length of the deposit. The values in millimetres are the channel widths $w$ and the values in degrees are the sidewall inclinations $\theta$. When the uncertainty bar is not visible, it is smaller than the symbol used. The straight line fits (by means of the least squares method) only the results of the simulations with $\theta = 27º$.

**Figure 10.** Plot of the reciprocal of mobility $\mu_A$ versus parameter $\chi$ where $L$ is set equal to the length of the flows. This length has been measured 0.58 sec after the gate removal in all flows (i.e., when they have already a fully developed shape of travel). Figures 9 and 10 illustrate the same numerical simulations. The values in millimetres are the channel widths $w$ and





the values in degree are the sidewall inclinations $\theta$. When the uncertainty bar is not visible, it is smaller than the symbol used. The straight line fits (by means of the least squares method) only the results of the simulations with $\theta = 27°$.

**Figure 11.** These illustrations portray particle agitation per unit of flow mass that increases as grain size $\delta$ increases (all the other features being the same). In this figure, the more agitated particles are represented by more distant polygons. The closer the position to the rough channel surfaces, the larger the agitation of the particles. These sketches are based on high speed video camera studies of granular flows (Cagnoli and Romano, 2012a, 2012b).

**Figure 12.** Plot of the reciprocal of mobility $\mu_A$ versus parameter $\beta$ of the same flow simulations illustrated in Figs. 9 and 10. The values in millimetres are the channel widths $w$ and the values in degrees are the sidewall inclinations $\theta$. When the uncertainty bar is not visible, it is smaller than the symbol used. Each straight line fits (by means of the least squares method) the results of simulations with the same channel width $w$. Only results of the simulations with $\theta = 27°$ are fitted by straight lines.





**Table 1.** Characteristics of channels and flows.

| Channel Width $w$ mm | Lateral Side Inclination $\theta$ | Total Flow Mass g | Grain Size $\delta$ mm |
|---|---|---|---|
| 6 | 27º | 8.9 | 1 |
| 6 | 27º | 13 | 1 |
| 6 | 27º | 26 | 1, 1.5, 2 |
| 10 | 27º | 29 | 1, 1.5, 2 |
| 16 | 27º | 33 | 1, 1.5, 2 |
| 26 | 27º | 39 | 1, 1.5, 2 |
| 26 | 27º | 93 | 2 |
| 16 | 27º | 68 | 2 |
| 6 | 19º | 36 | 1 |
| 6 | 41º | 17 | 1 |

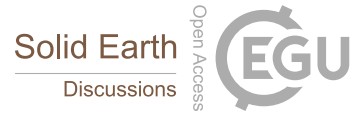

**Table 2.** Physical properties of materials.

|  | Particles | Channels[a] | Gates |
| --- | --- | --- | --- |
| Poisson's Ratio | 0.19 | 0.35 | 0.36 |
| Shear Modulus (Pa) | 2.38e+10 | 6.85e+09 | 25e+09 |
| Density (kg/m$^3$) | 2700 | 2580 | 2700 |

[a]Straight ramps and curved chutes have the same properties.



**Table 3.** Values of properties governing clast-clast, clast-channel and clast-gate interactions.

|  | Clast-Clast | Clast-Channel[a] | Clast-Gate |
| --- | --- | --- | --- |
| Coefficient of Restitution | 0.49 | 0.3 | 0.53 |
| Coefficient of Static Friction | 0.45 | 0.9 | 0.1 |
| Coefficient of Rolling Friction | 0.035 | 0.07 | 0.07 |

[a]Straight ramps and curved chutes have the same properties.




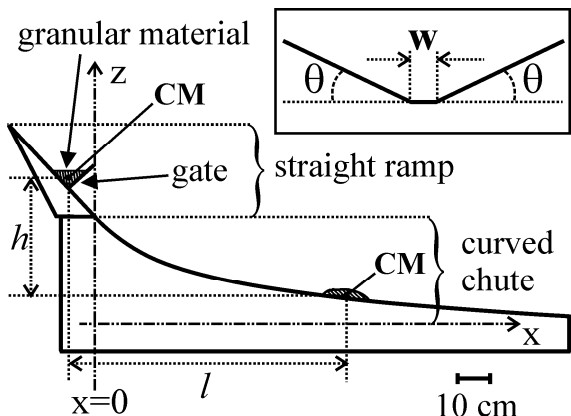

**Figure 1.** Longitudinal cross section of all the channels used in the numerical simulations. The inset illustrates the transversal cross section of both the straight ramps and the curved chutes where quantities $w$ (the channel width) and $\theta$ (the sidewall inclination) are shown. CM stands for center of mass.

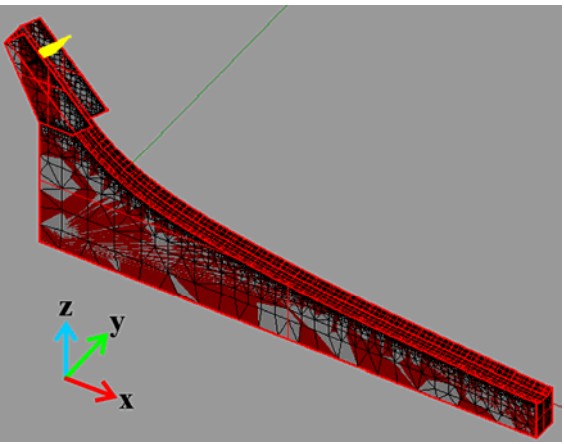

**Figure 2.** Example of a 3-D virtual channel generated by means of a CAD software (Rhinoceros) for the numerical simulations. The concave upward channel is shown in red and the gate in yellow.

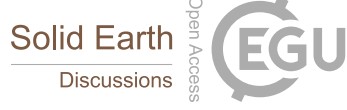

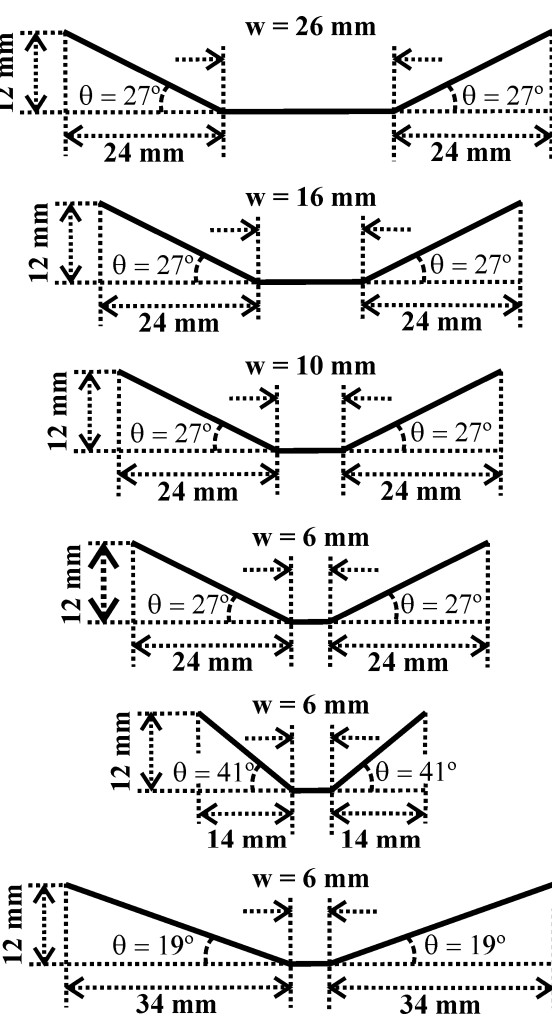

**Figure 3.** The different values of the channel width *w* and the sidewall inclination *θ* that are adopted in the numerical simulations.

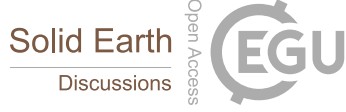



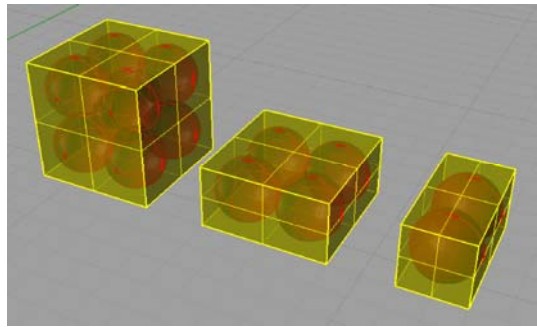

**Figure 4.** Shapes of the particles that are used in the numerical simulations. The inscribed clusters of spheres are also shown.

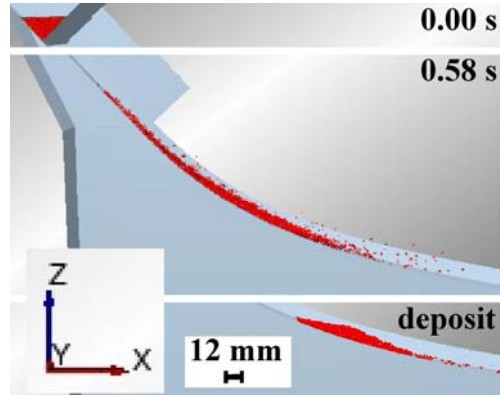

**Figure 5.** Cross sections illustrating three moments of a flow history. The frame at the top shows the granular material at rest behind the gate before release. The frame in the middle shows the fully developed shape of travel of the granular flow. The lower frame shows the final deposit. In this example the granular mass is 93 g, the grain size is 2 mm, $w$ is 26 mm and $\theta$ is 10    27º. The values in seconds are the time measured after the gate removal.



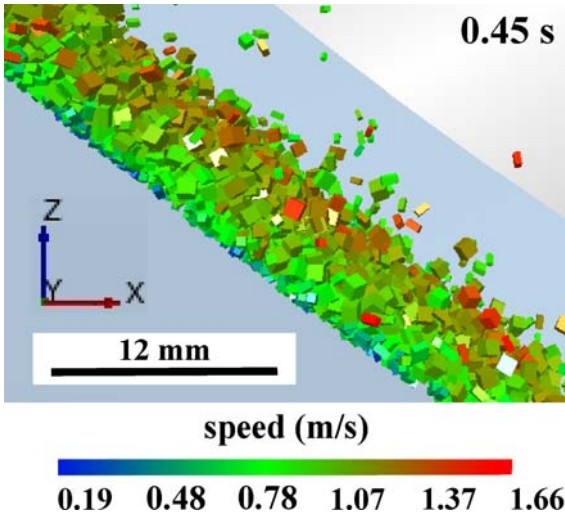

**Figure 6.** Side view of a granular flow. Particle speed decreases toward the subsurface along its normal. In this example the granular mass is 26 g, the grain size is 1 mm, $w$ is 6 mm and $\theta$ is 27°. The time in seconds is measured after the gate removal.





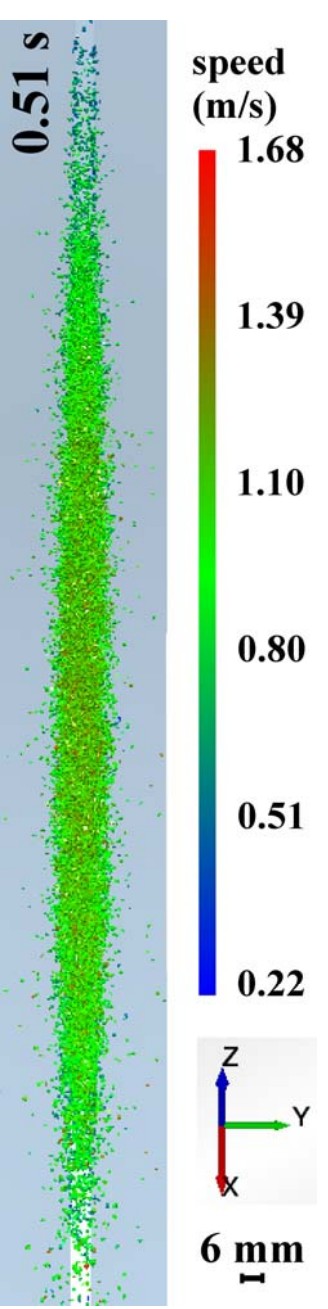

**Figure 7.** Fully developed shape of travel of a granular flow seen from above 0.51 s after the gate removal. In this example, the granular mass is 26 g, the grain size is 1 mm, *w* is 6 mm and *θ* is 27º.





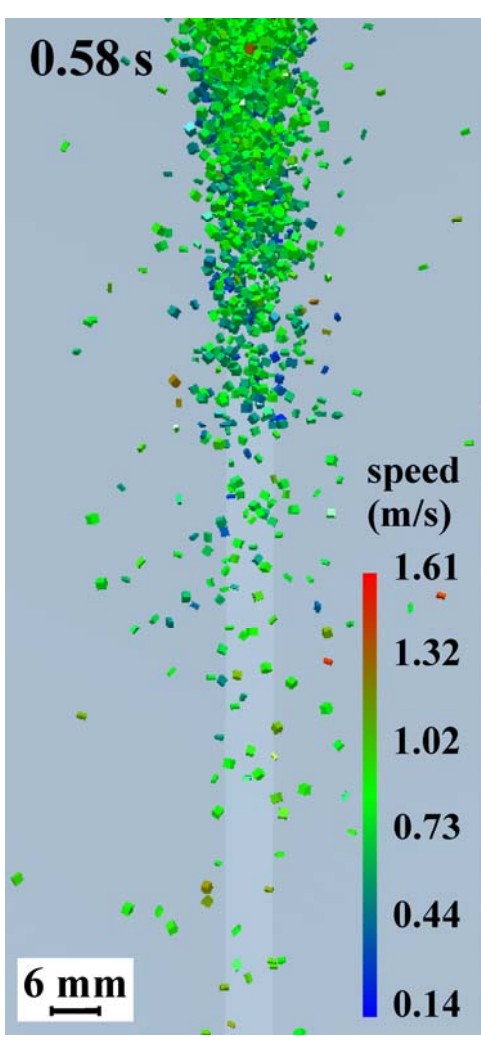

**Figure 8.** Flow front seen from above 0.58 s after the gate removal. The saltating particles of the distal distribution in front of the dense granular flow are clearly identifiable. In this example the granular mass is 26 g, the grain size is 1 mm, *w* is 6 mm and $\theta$ is 27º.





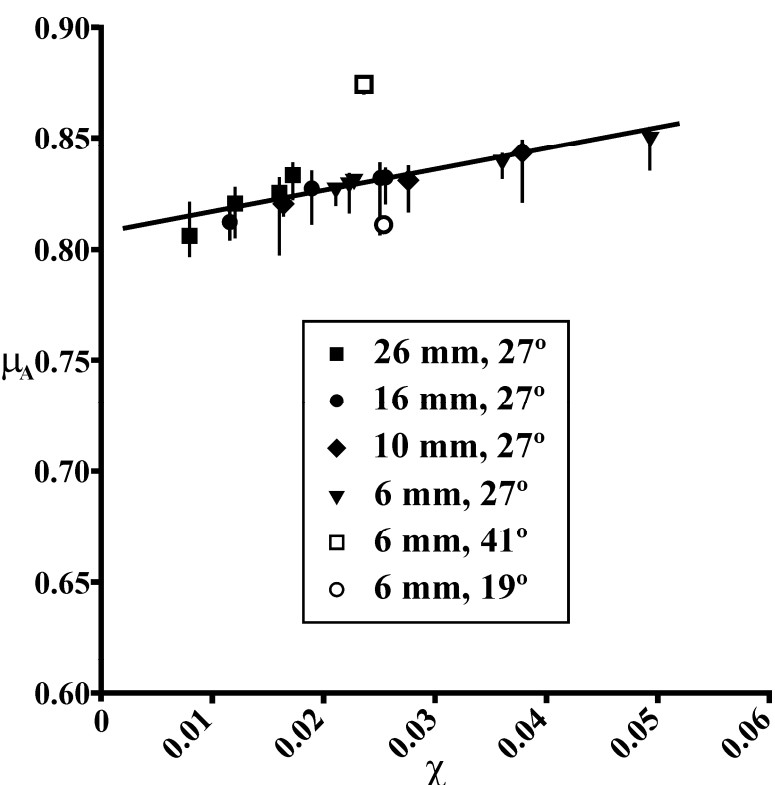

**Figure 9.** Plot of the reciprocal of mobility $\mu_A$ versus parameter $\chi$ where $L$ is set equal to the length of the deposit. The values in millimetres are the channel widths $w$ and the values in degrees are the sidewall inclinations $\theta$. When the uncertainty bar is not visible, it is smaller than the symbol used. The straight line fits (by means of the least squares method) only the results of the simulations with $\theta = 27°$.





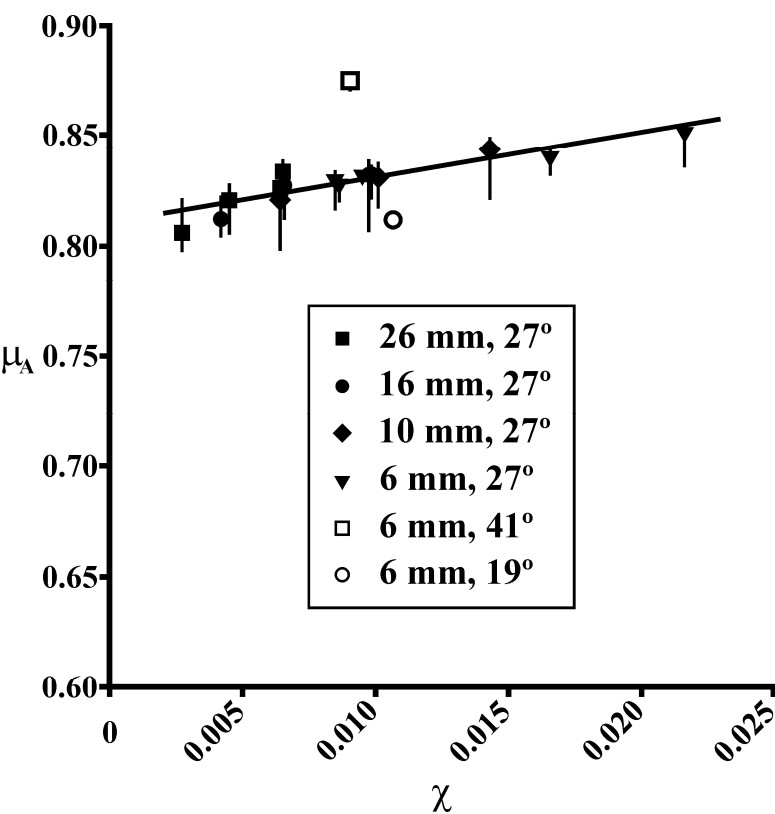

**Figure 10.** Plot of the reciprocal of mobility $\mu_A$ versus parameter $\chi$ where $L$ is set equal to the length of the flows. This length has been measured 0.58 sec after the gate removal in all flows when they have already a fully developed shape of travel. Figures 9 and 10 illustrate the same numerical simulations. The values in millimetres are the channel widths $w$ and the values in degree are the sidewall inclinations $\theta$. When the uncertainty bar is not visible, it is smaller than the symbol used. The straight line fits (by means of the least squares method) only the results of the simulations with $\theta = 27°$.



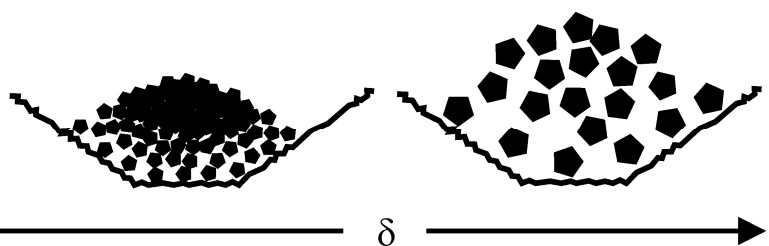

**Figure 11.** These illustrations portray particle agitation per unit of flow mass that increases as grain size $\delta$ increases (all the other features being the same). In this figure, the more agitated particles are represented by more distant polygons. The closer the position to the rough channel surfaces, the larger the agitation of the particles. These sketches are based on high speed video camera studies of granular flows (Cagnoli and Romano, 2012a, 2012b).





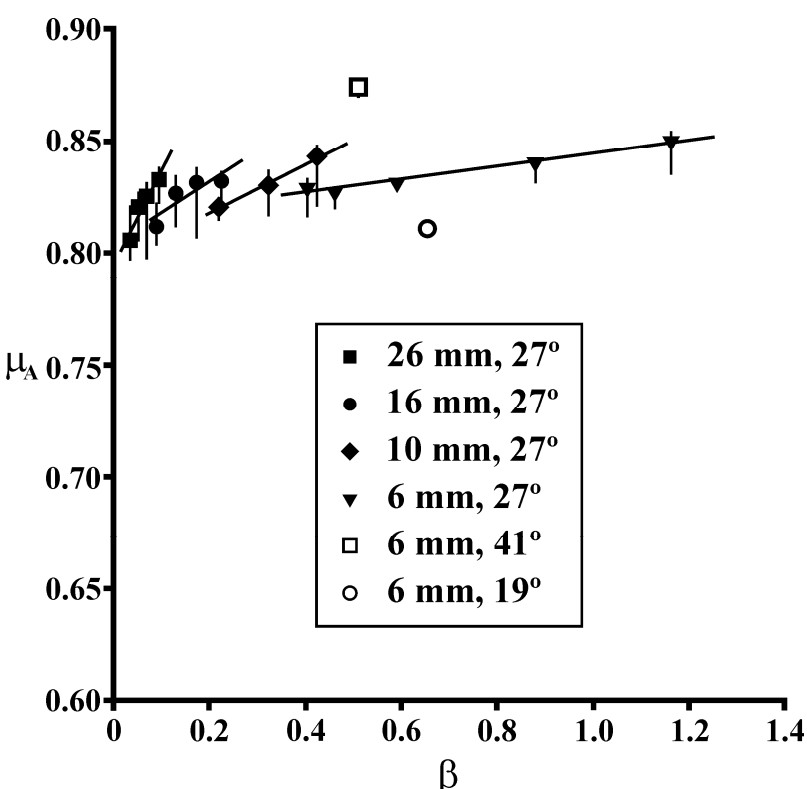

**Figure 12.** Plot of the reciprocal of mobility $\mu_A$ versus parameter $\beta$ of the same flow simulations illustrated in Figs. 9 and 10. The values in millimetres are the channel widths $w$ and the values in degrees are the sidewall inclinations $\theta$. When the uncertainty bar is not visible, it is smaller than the symbol used. Each straight line fits (by means of the least squares method) the results of simulations with the same channel width $w$. Only results of the simulations with $\theta = 27°$ are fitted by straight lines.