# Peer review of "Combined effects of grain size, flow volume and channel width on geophysical flow mobility: 3-D discrete element modeling of dry and dense flows of angular rock fragments"

_Solid Earth, 2016_

## Referee Comment (RC1) · M. Zhang (Referee) · 2 Aug 2016

This study investigated the impact of several factors on mobility of granular flow, which is a hot topic in mobility of rock avalanche. Several new findings were obtained through numerical simulations and physical simulations previously conducted by authors. However, there are still some questions needing authors to answer or revise. (1) Lines 20-21, Page 4: When I first read granular flow mass in this paper, many questions arose in my mind included: What does this mean? How to determine it? Why the authors

use all those masses in the simulation? Why do not the authors use particle number or other parameters to quantify of total particles used in simulation? I understood until I finished reading the paper. Could authors please simply explain those questions when it first appears? (2) How to determine the properties between particles, particles and channels, particle and gates in this paper? When we do numerical simulation using discrete element method, one of the most important procedures is to determine the micro-parameters of and between elements. In this research, the authors directly gave the parameters without explanation. (3) This research used centre of mass of deposit to calculate the mobility of granular flow in the numerical and physical simulations. However, it is very difficult to determine the centre of deposit, especially in physical simulation and in granular deposits of a real rock avalanche. Could the authors please explain how to determine centre of the deposit in their physical simulations? (4) Page 9, Lines 23∼25: "The collapse along a single straight line of all the data points of the simulations with $\theta$ = 27° confirms that, in Figs. 9 and 10, only the variables considered in Eq. (19) have values that vary and, consequently, determine the observed different mobility of the centre of mass of the different flows." âŚă Not only the variables considered in the equation (19) determine the mobility of granular flow, many other variables not considered also have influence, which were actually the constant in this research. âŚą About angle of sidewall, the authors only used 19 ° and 41 °, which is too few to find the fitness. Furthermore, they did not try to fit the three data points with width of 6 mm and three different angles of 19 °, 27 ° and 41 °. Therefore, it is not reasonable to exclude angle of sidewall as a factor affecting the mobility. (5) Page 10, Lines 25∼ 28: Authors should add the latest research "Zhang, M., Yin, Y., McSaveney, M. (2016) Dynamics of the 2008 earthquake-triggered Wenjiagou Creek rock avalanche, Qingping, Sichuan, China ", which also drew the conclusion the mobility of granular flow increases with finer grain size. (6) About impact of the volume of granular flow on mobility, the conclusion in this research is much different from our generally accepted one that mobility increases with increased volume of the granular flow. Even if this research used centre of the deposit to calculate the u/l, the conclusion is different from

the research conducted by several scientists on mobility of rock avalanches (Davies et al., 1999). (7) About impact of channel width, the conclusion in this research is contrary to the statistic results on rock avalanches conducted by Nicoletti and Sorriso-valvo (1991). (8) The authors did not consider grain fragmentation during movement in their physical numerical simulations, which plays a very important role in the mobility of granular mobility. Actually, many scientists (Davies and McSaveney, 2009; De Blasio and Crosta, 2014, 2015) reached the conclusion that physical simulation cannot repeat the high mobility of granular debris flow because it is not able to simulate the pervasive grain fragmentation during movement. Could the authors please explain the reason and the impact that the grain fragmentation was not considered in this research? Another two corrections: (1) Caption of Fig. 9. Delete "The alues in millimetres are the channel widths w and the values in degrees are the sidewall inclinations $\theta$". (2) Caption of Fig. 10. Delete "The alues in millimetres are the channel widths w and the values in degrees are the sidewall inclinations $\theta$".

---

## Author Comment (AC1) · 11 Aug 2016

We would like to thank Ming Zhang for his good comments that are of general interests. The answers are provided here below. The Line and Page numbers refer to the new version of the manuscript which is attached.

(1) COMMENT: Lines 20-21, Page 4: When I first read granular flow mass in this paper, many questions arose in my mind included: What does this mean? How to

determine it? Why the authors use all those masses in the simulation? Why do not the authors use particle number or other parameters to quantify of total particles used in simulation? I understood until I finished reading the paper. Could authors please simply explain those questions when it first appears?

(1) ANSWER:

> The mass is an important quantity because, since the particle density is a constant, it is proportional to flow volume (Page 9, Lines 18-19) and the rock avalanches are commonly described in the field by means of their volumes. The number of particles is not a practical quantity to measure the size of a rock avalanche because it is very hard (if not impossible) to assess it in the field.

> In scaling analysis, the actual mass values that are used are not relevant provided that the range of mass values is relatively wide so that the effect of volume on the collapse of the scaling parameters along a single straight line can be evaluated. We test here a relatively large number of mass values because the volume effect on mobility is considered important in the literature (see your comment 6). We have now recapped these concepts on Page 4, Lines 29-33 (after the introduction of the mass values).

(2) COMMENT: How to determine the properties between particles, particles and channels, particle and gates in this paper? When we do numerical simulation using discrete element method, one of the most important procedures is to determine the micro-parameters of and between elements. In this research, the authors directly gave the parameters without explanation.

(2) ANSWER:

> The values of the physical properties of materials and their interactions are illustrated in Tables 2 and 3. In Section 2.1 we explain that these values represent interactions among igneous rock fragments which travel on a subsurface made of soil (Peng, 2000). These are the same values that we have adopted in the simulations by Cagnoli and

[Figure]

[Figure]

Piersanti (2015). They were adopted because, with these values, it has been possible to obtain the same relative flow mobility of the different flows as that observed in the laboratory experiments by Cagnoli and Romano (2010 and 2012a). We now recall this on Page 5, Lines 9-11.

(3) COMMENT: This research used centre of mass of deposit to calculate the mobility of granular flow in the numerical and physical simulations. However, it is very difficult to determine the centre of deposit, especially in physical simulation and in granular deposits of a real rock avalanche. Could the authors please explain how to determine centre of the deposit in their physical simulations?

(3) ANSWER:

> Any CAD software is able to compute the position of the center of mass of virtually any 3D solid (or collection of solids) no matter its (or their) shape. In numerical simulations, the position in space of all the particles is known at any time (from behind the gate to the final deposition). When these positions are imported in a CAD software, it is straightforward to compute the position of their center of mass. On Page 8, Lines 6-7, we now state that we have used a CAD software (Rhinoceros) to compute the positions of the centers of mass.

> In the laboratory experiments (and this can be repeated in the field), we constructed three-dimensional representations (i.e., the shape in 3D) of the final deposits of the rock fragments which were then imported in a CAD software for the calculations of the position of their center of mass. The same applies to the granular masses behind the gate before release. This is explained in Cagnoli and Romano, JVGR, 2010a and Cagnoli and Romano, JGR, 2012a.

(4) COMMENT: Page 9, Lines 23_25: "The collapse along a single straight line of all the data points of the simulations with $\theta$=27° confirms that, in Figs. 9 and 10, only the variables considered in Eq. (19) have values that vary and, consequently, determine the observed different mobility of the centre of mass of the different flows." â′S ËŸa Not

only the variables considered in the equation (19) determine the mobility of granular flow, many other variables not considered also have influence, which were actually the constant in this research. â′SaËŻ About angle of sidewall, the authors only used 19° and 41°, which is too few to find the fitness. Furthermore, they did not try to fit the three data points with width of 6 mm and three different angles of 19°, 27° and 41°. Therefore, it is not reasonable to exclude angle of sidewall as a factor affecting the mobility.

(4) ANSWER:

> In general, there are several variables which affect granular flow mobility. Some of them are discussed in Section 5.4 to provide a useful background. In our numerical simulations only grain size, flow volume and channel width have values which vary. It is correct to vary at the same time (i.e., in the same set of simulations and experiments) the values of these three variables because these quantities can be bundled into a single scaling parameter (Eq. 19).

> However two additional ancillary simulations have been carried out also with two different $\theta$ values. Our numerical simulations show that the sidewall inclination $\theta$ does affect mobility. At the end of Section 4 (Page 11, Lines 1-2), we actually write that Figs 9 and 10 show that the larger the sidewall inclination $\theta$, the larger the values of $\mu$A. We say also that simulations with different $\theta$ values plot along different straight lines.

> In Figs 9 and 10, eq. (18) fits only the data with $\theta$=27° (i.e., the quantities not in eq. (19) are constant). This is so because the angle $\theta$ does no enter parameter $\chi$ (eq. (19)). Angle $\theta$ cannot enter parameter $\chi$ because $\theta$ is itself another dimensionless scaling parameter and it does not make sense to multiply or divide these two scaling parameters among themselves. We now state that $\theta$ is an independent dimensionless scaling parameter on Page 11, Line 1.

> In Fig. 9 and 10, the data point of the simulation with w=6mm, $\delta$=1mm and $\theta$=27° is the triangle with the smaller $\mu$A value. You can see that the three data points you
mention plot roughly along a line that is almost vertical. However, three points are not enough to be sure about the shape of the functional relation, if any. Since $\theta$ is an independent scaling parameter, chances are that the three points you mention are not fitted by a single curve. We are also not much interested in $\theta$ values here because we want to verify the laboratory results where $\theta$ does not vary because it cannot enter $\chi$.

(5) COMMENT: Page 10, Lines 25_28: Authors should add the latest research "Zhang, M., Yin, Y., McSaveney, M. (2016) Dynamics of the 2008 earthquake-triggered Wenjiagou Creek rock avalanche, Qingping, Sichuan, China ", which also drew the conclusion the mobility of granular flow increases with finer grain size.

(5) ANSWER:

> We have now cited this interesting paper on Page 12, Line 10.

(6) COMMENT: About impact of the volume of granular flow on mobility, the conclusion in this research is much different from our generally accepted one that mobility increases with increased volume of the granular flow. Even if this research used centre of the deposit to calculate the u/l, the conclusion is different from the research conducted by several scientists on mobility of rock avalanches (Davies et al., 1999).

(6) ANSWER:

> Our conclusions do not really contradict what previously said about the volume effect. We, instead, highlight an important aspect of a multifaceted issue.

> We believe that the larger the flow volume, the longer the longitudinal spreading of the deposits as suggested by Davies (1982) and reported by other authors (D'Agostino et al., 2010) and this is so, in a channel for example, because the planimetric area inundated by a flow is proportional to a power of the flow volume (Griswold and Iverson, 2008). This generates an inverse correlation between flow volume and mobility when this mobility is measured by considering the front of the deposit. This relation is not particularly useful, though, because the correlation coefficient can be quite small

(please see, for example, Fig. 2c in Nicoletti and Sorriso-Valvo, 1991).

> However we have proved by two independent methods (laboratory experiments and numerical simulations) that when the mobility is measured by considering the center of mass, the larger the flow volume, the smaller the mobility of the center of mass. This is the same phenomenon observed independently by Okura et al., (2000) in their much larger experimental apparatus. This is our view reported on Page 2, Lines 19-27.

(7) COMMENT: About impact of channel width, the conclusion in this research is contrary to the statistic results on rock avalanches conducted by Nicoletti and Sorrisovalvo (1991).

(7) ANSWER:

> In our paper, all our flows are channeled and we compare their mobility in channels with different width. The narrower the channel, the less mobile the center of mass. This is so because the deposit propagates backward during its formation and, the narrower the channel, the longer the backward propagation (it is also possible that in a narrower channel the retarding effect of the sidewalls is relatively more important). Nicoletti and Sorriso-Salvo (1991), on the other hand, do not compare the mobility of flows in channels with different width. They compare channeled flows with flows that have no lateral constraints.

> A different mechanism explains the differences that are observed when comparing the mobility of channeled flows with the mobility of flows without lateral constraints. In this case, the flows without lateral constraints tend to be less mobile (Nicoletti and Sorriso-Valvo, 1991). This is due to the fact that a flow without lateral constraints spreads laterally, whereas the entire momentum of a channeled flow is spent along one single direction: that of the channel. We have added this explanation on Page 14, Lines 8-15.

(8) COMMENT: The authors did not consider grain fragmentation during movement in

their physical numerical simulations, which plays a very important role in the mobility of granular mobility. Actually, many scientists (Davies and McSaveney, 2009; De Blasio and Crosta, 2014, 2015) reached the conclusion that physical simulation cannot repeat the high mobility of granular debris flow because it is not able to simulate the pervasive grain fragmentation during movement. Could the authors please explain the reason and the impact that the grain fragmentation was not considered in this research?

(8) ANSWER:

> We thank the reviewer for this comment that gives us the opportunity to explain how we work. In our paper, we investigate the effect of grain size on flow mobility. This has been carried out by holding constant the value of grain size during flow motion to prevent other phenomena from interfering with the grain size effect under study. In laboratory experiments (our numerical simulations are also meant to verify independently our laboratory results), it is good practice to study only one phenomenon at the time. We believe that good physical simulations should not reproduce the entire complexity of natural flows because a) this would be impossible and b) different phenomena would interfere with one another preventing the researcher from reaching any solid conclusion on the effect of any single quantity.

> In any case, we believe that, in nature, particle-particle and particle-boundary interactions cause particle fragmentation during flow motion. We observed this phenomenon in stationary granular flows during laboratory experiments (please see Fig 7 in Cagnoli and Manga, 2004). In rock avalanches, there is field evidence that grain size decreases as travel distance increases and that flows with long runouts are associated with a reduced grain size (Davies and McSaveney, 2009; Zhang et al., 2016). In particular, Davies and McSaveney (2009) advance a theory in which they suggest that the fragmentation process in itself can increase flow mobility. However, our research (by means of both laboratory experiments and numerical simulations) also demonstrates that, when rock fragmentation does not occur, the mere presence of fine grain size is conducive to a reduced energy dissipation of the granular flows. We have now added

this explanation on Page 12, Lines 5-14.

(9) COMMENT: Another two corrections: (1) Caption of Fig. 9. Delete "The values in millimetres are the channel widths w and the values in degrees are the sidewall inclinations $\theta$". (2) Caption of Fig. 10. Delete "The values in millimetres are the channel widths w and the values in degrees are the sidewall inclinations $\theta$".

(9) ANSWER:

> These sentences are necessary to explain the content of Figs. 9 and 10.

Please also note the supplement to this comment:
http://www.solid-earth-discuss.net/se-2016-79/se-2016-79-AC1-supplement.pdf

**Supplement:**

[revised manuscript text omitted]

The numerical flows are dry and consist of particles with three different shapes. We use particles with a cubic shape, half a cubic shape and a quarter of a cubic shape (Fig. 4). These polyhedrons represent an equant, an oblate and a prolate particle, respectively. Non-spherical particles are preferred because when interacting among themselves and with the boundary surfaces, their energy dissipation mechanism (due to collisions and attrition) is comparable with that of natural fragments since they are both angular (Cagnoli and Romano, 2012a; Mead and Cleary, 2015). The proportion of each particle shape is always the same in all flows irrespective of grain size, flow mass or channel features: the equant particles are 38%, the oblate particles 22% and the prolate particles 40% of the flow mass. This generates more realistic flows than those with particles of only one shape, because natural geophysical flows contain particles with different shapes. We have shown that flows with different proportions of particle shapes dissipate different amounts of energy per unit of travel distance (Cagnoli and Piersanti, 2015). For this reason, it is important that the comparison of mobility is carried out among flows with the same proportions of particle shapes. Concerning grain size, we adopt geometrically similar polyhedrons whose longer edges can be 1, 1.5, or 2 mm in length. Only one grain size is used in each granular flow.

Table 1 illustrates the combinations of the values of the channel width $w$, the lateral side inclination $\theta$, the grain size $\delta$ and the total granular flow mass adopted in the numerical simulations. In the channel with $w = 6$ mm and $\theta = 27º$, we use total granular flow masses equal to 8.9, 13 and 26 g. This total mass increases in wider channels with $\theta = 27º$: 29 g when $w = 10$ mm, 33 g and 68 g when $w = 16$ mm, 39 g and 93 g when $w = 26$ mm. The flow mobility of simulations with sidewall inclinations $\theta$ equal to 19º and 41º (where $w = 6$ mm and $\delta = 1$ mm) is meant to be compared with the mobility obtained with $\theta = 27º$ and the same channel width $w$ and the same grain size $\delta$. In the simulations with $\theta$ equal to 19º and 41º, the total granular flow mass is 36 and 17 g, respectively. Since particle density is a constant, the granular flow mass is proportional to flow volume whose effect on mobility we are interested in. The actual granular mass values we use are not important in a scaling analysis provided that the range of values is relatively wide so that the effect of volume on the collapse of the scaling parameters along a single line can be evaluated. Here we test a relatively large range of total granular mass values because the effect of volume on mobility is controversial.

Table 2 shows the values of the physical properties of particles, channels and gates adopted in the numerical simulations. The properties of the particles are those of an igneous rock, the properties of the channels are those of clay and the properties of the gates are those of aluminum. Table 3 shows the values of the properties that govern particle-particle, particle-channel and particle-gate interactions. These values indicate that we are simulating flows of rock fragments that travel on a subsurface made of soil (Peng, 2000). The angle of internal friction of the granular material is not explicitly mentioned by the model, but, this important property is determined by the shapes of the particles. Our numerical simulations pertain to angular fragments. The surface of the gates has a very small friction value to avoid disturbing the granular material when the gates are removed. The roughness of the chute and ramp surfaces is identical and it is everywhere significantly smaller than the smallest grain size we use. The values shown in Tables 2 and 3 are the same as those adopted in the numerical simulations by Cagnoli and Piersanti (2015) because, with these values, it has been possible to obtain the same relative mobility of the different flows as that observed in the laboratory by Cagnoli and Romano (2010, 2012a).

In our software (EDEM), we need to locate the particles at time zero. For this purpose, we generate, behind the gate, 3-D abstract spaces that are filled (before the gate is removed) with particles in random position and without interpenetration. These abstract spaces do not represent any real material object. We use abstract spaces whose volumes are proportional to the granular masses so that the same compaction and density of the granular material behind the gate before release is obtained. We have shown that the larger the initial compaction of the granular material behind the gate before release, the larger the mobility of the centre of mass of the granular flow (Cagnoli and Piersanti, 2015). Therefore it is important that the comparison of mobility is carried out among granular masses that before release have the same initial compaction. Within our abstract spaces, the particles are in contact with one another and the granular masses have a bulk average density equal to $721\pm14$ kg/m$^3$. The density of the particle material is 2700 kg/m$^3$ (Table 2). In the numerical simulations we use granular masses that are relatively small so that the computer processing times are manageable. The flows with the largest number of particles (~34000) are those with a total mass equal to 39 g and a grain size equal to 1 mm. The flows with the smallest number of particles (~2800) are those with a total mass equal to 26 g and a grain size equal to 2 mm.

**2.2. Contact Model of the Numerical Simulations**

Our 3-D discrete element modeling has been carried out by using the software EDEM developed by DEM Solutions (www.dem-solutions.com). The approach that EDEM adopts when dealing with particles is twofold. On one hand, it adopts the mass, volume and moment of inertia of the polyhedrons we have chosen. On the other hand, it uses clusters of spheres inscribed within the polyhedrons (Fig. 4) to estimate impact forces during particle collisions. These forces are a function of sphere overlaps. Experience has shown that clusters of spheres are an effective method to model complex particle shapes with a good degree of approximation (DEM Solutions, 2014). This method allows good computing performance because the contact detection algorithm for clusters of spheres is more efficient than that for polyhedrons. For particle collisions, the model computes normal and tangential forces, their damping components and the tangential and rolling friction forces (DEM Solutions, 2014).

The normal force (Hertz, 1882) is

$$F_n = \frac{4}{3} E^* \sqrt{R^*} \lambda_n^{3/2} , \tag{2}$$

where $\lambda_n$ is the normal overlap, $E^*$ is the equivalent Young's modulus and $R^*$ is the equivalent radius that are defined as follows:

$$\frac{1}{E^*} = \frac{(1 - v_i^2)}{E_i} + \frac{(1 - v_j^2)}{E_j} \tag{3}$$

and

$$\frac{1}{R^*} = \frac{1}{R_i} + \frac{1}{R_j} , \tag{4}$$

respectively. Here, $E$ and $v$ are the Young's moduli and the Poisson's ratios, respectively, of the interacting elements i and j (polyhedrons, channel or gate). $R_i$ and $R_j$ are the radii of the interacting spheres of the interacting polyhedrons i and j. When

10 one of the two interacting elements is not a particle, the equivalent radius is equal to the radius of the interacting sphere of the polyhedron.

The tangential force (Mindlin, 1949; Mindlin and Deresiewicz, 1953) is

$$F_t = -S_t \lambda_t , \tag{5}$$

where $\lambda_t$ is the tangential overlap and $S_t$ is the tangential stiffness which is a function of the equivalent shear modulus $G^*$.

15 The stiffness is

$$S_t = 8 G^* \sqrt{R^* \lambda_n} \tag{6}$$

and the equivalent shear modulus is

$$\frac{1}{G^*} = \frac{(2 - v_i)}{G_i} + \frac{(2 - v_j)}{G_j} , \tag{7}$$

where $G_i$ and $G_j$ are the shear moduli of the interacting elements. The tangential force is limited by Coulomb's friction which

20 is equal to

$$\mu_s F_n , \tag{8}$$

where $\mu_s$ is the coefficient of static friction (Cundall and Strack, 1979).

The normal and tangential damping components (Tsuji et al., 1992) are

$$F_n^d = -2\sqrt{\frac{5}{6}}\,\varepsilon\sqrt{S_n m^*}\,u_n^{rel} \tag{9}$$

and

$$F_t^d = -2\sqrt{\frac{5}{6}}\,\varepsilon\sqrt{S_t m^*}\,u_t^{rel}, \tag{10}$$

respectively. In Eqs. (9) and (10),

5  $$\varepsilon = \frac{\ln e}{\sqrt{\ln^2 e + \pi^2}}, \tag{11}$$

$$m^* = \left(\frac{1}{m_i} + \frac{1}{m_j}\right)^{-1} \tag{12}$$

and

$$S_n = 2E^*\sqrt{R^* \lambda_n}, \tag{13}$$

where $m^*$ is the equivalent mass, $m_i$ and $m_j$ are the masses of the interacting elements (polyhedrons, channels or gates), $e$ is
10  the coefficient of restitution, $S_n$ is the normal stiffness and the $u^{rel}$ values are the normal (subscript n) and tangential (subscript t) components of the relative velocity.

The rolling friction (Sakaguchi et al., 1993) is accounted for by applying a torque

$$\tau_i = -\mu_r F_n d_i \omega_i \tag{14}$$

to the contacting surfaces. Here $\mu_r$ is the coefficient of rolling friction, $d_i$ is the distance from the centre of mass of the
15  polyhedron to the contact point (where the contact point is defined in the middle of the overlap) and $\omega_i$ is the unit angular velocity vector of the particle at the contact point. This torque is calculated independently for each polyhedron.

**3 Scaling**

We measure the reciprocal of mobility of a flow by using the apparent coefficient of friction

$$\mu_A = \frac{h}{l}, \tag{15}$$

where $h$ is the vertical drop of the centre of mass of the granular material and $l$ is its horizontal distance of travel. Distances $h$ and $l$ are measured from the position of the centre of mass of the granular samples at rest behind the gate before release to the position of the centre of mass of the final deposits (Fig. 1). In an energy dissipation (i.e., mobility) analysis, the knowledge of the whereabouts of the centre of mass is a necessity because the centre of mass is the only point that can be
5   used as a proxy for the entire flow since it moves as though the total mass of all the particles were concentrated there and all the external forces were applied there. The positions in space of the centres of mass have been computed here by means of a 3-D CAD software (Rhinoceros). Our parameter, thus, differs from the so-called Heim coefficient that considers the vertical and horizontal distances between the highest elevation point of the failed landscape feature (a mountain slope or a volcanic dome, for example) and the most distal position of the front of the deposit.

10      In both the laboratory experiments (Cagnoli and Romano, 2012a) and the numerical simulations (here and in Cagnoli and Piersanti, 2015), the deposited granular material consists of two portions: a more proximal heap that is much more elongated than thick (the deposit of the flow proper) and a more distal distribution of individual fragments. The distal distribution is formed by fragments, which, bouncing within the chute, traveled individually without interacting and are not part of the flow proper. Flows and distal distributions have different movement and depositional mechanisms and they must be considered
15   separately. Here we study the flow proper. Therefore, the value of $\mu_A$ is computed for the final deposit of the flow proper, which consists of all the particles on the chute that are in contact with one another. The particles of the distal distribution are those that are not in contact with one another. However, for comparison, we show also the value of $\mu_A$ computed for the combined particles of the final deposit of the flow proper and the distal distribution together.

[revised manuscript text omitted]

5    In our paper, we investigate the effect of grain size on flow mobility. This has been carried out by holding constant the value of grain size during flow motion to prevent other phenomena from interfering with the grain size effect under study. However, we believe that, in nature, particle-particle and particle-boundary interactions cause particle fragmentation during flow motion. We observed this phenomenon in stationary granular flows during laboratory experiments (Cagnoli and Manga, 2004). In rock avalanches, there is field evidence that grain size decreases as travel distance increases and that flows with

10   long runouts are associated with a reduced grain size (Davies and McSaveney, 2009; Zhang et al., 2016). In particular, Davies and McSaveney (2009) advance a theory in which they suggest that the fragmentation process in itself can increase flow mobility. However, our research (by means of both laboratory experiments and numerical simulations) demonstrates that, even when rock fragmentation does not occur, the mere presence of fine grain size is conducive to a reduced energy dissipation of the granular flows.

[revised manuscript text omitted]

5    channel width, the grain size and the flow volume). Quantity $L$ has, however, been introduced in Eq. (16) because the knowledge alone of $V$ and $w$ does not suffice to reckon the extent of the longitudinal spread of the particles within the channel since, in this equation, the thickness of the flow is missing.

In our simulations, all our flows are channelled and we compare their mobility in channels with different widths. The narrower the channel, the less mobile the centre of mass. This is so because the deposit propagates backward during its
10    formation and, the narrower the channel, the longer the backward propagation (it is also possible that in a narrower channel the retarding effect of the sidewalls is relatively more important). However, a different mechanism explains the different behaviours that are observed when comparing the mobility of channelled flows and the mobility of flows without lateral constraints. In this case, the flows without lateral constraints tend to be less mobile (Nicoletti and Sorriso-Valvo, 1991). This is due to the fact that a flow without lateral constraints spreads laterally, whereas the entire momentum of a channelled flow
15    is spent along a single direction: that of the channel.

**5.4 Other Quantities Affecting Flow Mobility**

[revised manuscript text omitted]

---

## Referee Comment (RC2) · M. Zhang (Referee) · 23 Aug 2016

Most of the comments have been revised or revised. But I still have two questions as follows: (1) I know you can obtain the gravity center through importing the three-dimensional shape into CAD. But how do you obtain the three-dimensional shape of the deposit especially in a real rock avalanche with volume of more than 1 million m3? Actually some scientists including the reviewer conducted the statistics on relationship between u/l and possible influencing factors, but they used top point of the scar and

distal point of the deposit to calculate the u/l because it is hard to obtain distribution of deposits of rock avalanches since it is complicated. (2) In answer (6), it not true to say "This generates an inverse correlation between flow volume and mobility when this mobility is measured by considering the front of the deposit." Actually, they obtained the inverse relationship between volume and u/l if calculated using the front of the deposit.

---

## Referee Comment (RC3) · M. Zhang (Referee) · 23 Aug 2016

I think this MS could be published since the authors have revised and explained all the questions.

---

## Author Comment (AC2) · 23 Aug 2016

Comments and answers are provided here below. The Line and Page numbers refer to the new version of the manuscript.

(1) COMMENT: I know you can obtain the gravity center through importing the three dimensional shape into CAD. But how do you obtain the three-dimensional shape of the deposit especially in a real rock avalanche with volume of more than 1 million m3?

[Figure]

Actually some scientists including the reviewer conducted the statistics on relationship between u/l and possible influencing factors, but they used top point of the scar and distal point of the deposit to calculate the u/l because it is hard to obtain distribution of deposits of rock avalanches since it is complicated.

(1) ANSWER:

> The best way to obtain the 3D shape of the rock avalanche in the field is to compute the difference between the digital elevation model of the ground surface after the landslide and that before the landslide. A digital elevation model can be obtained by means of satellite data (radar data for example) or a more traditional land survey (total station).

(2) COMMENT: In answer (6), it not true to say "This generates an inverse correlation between flow volume and mobility when this mobility is measured by considering the front of the deposit." Actually, they obtained the inverse relationship between volume and u/l if calculated using the front of the deposit.

(2) ANSWER:

> Yes, sorry, this is only a typo in the rebuttal letter. In answer 6, we are of course talking about the inverse correlation between flow volume and the apparent coefficient of friction u/l, such as that shown by Scheidegger 1973 (we have attached his figures). We have now cited Scheidegger 1973 on Page 8, Line 9. Thank you.

Please also note the supplement to this comment:
http://www.solid-earth-discuss.net/se-2016-79/se-2016-79-AC2-supplement.pdf

**Supplement:**

**Scheidegger,1973:**

[Figure]

Fig. 1. Geometry of a landslide

Geometrie eines Bergsturzes

Géometrie d'un éboulement

**Scheidegger,1973:**

[Figure]

Fig. 2. Correlation between landslide volume V (in m³) and the coefficient of friction $f$.

---

## Short Comment (SC1) · 22 Sep 2016

Based on an initial scan of the manuscript I noted distinct similarity and repetition of results obtained in your 2015 JGR paper. Grain size and flow volume were already studied in detail there, so I do not see the point to repeat too large quantities from that study. Besides large quantities of text (19% of the text resembles your 2015 study according to the similarity report), I also see several figures and tables are exactly the same (e.g., Fig. 1,2, and 4), while all other figures are also very similar to your JGR

2015 paper. In my opinion such large amounts of overlap is surely not warranted in scientific publications.

Additionally, it seems that in your "new" scaling parameter you only replace the length scale in the denominator from width * width to width * length. Is this correct? If so, I would acknowledge that within the paper and tune down the formulation. I would say it is a (small) update of a scaling parameter.

---

## Author Comment (AC3) · 28 Sep 2016

We would like to thank Ylona van Dinther for her comments and we would like to congratulate her on the birth of her new baby and wish her all the best.

(1) COMMENT Based on an initial scan of the manuscript I noted distinct similarity and repetitions of results obtained in your 2015 JGR paper. Grain size and flow volume were already studied in detail there, so I do not see the point to repeat too large quantities from that study. Besides large quantities of text (19% of the text resembles your 2015 study according to the similarity report), I also see several figures and tables are exactly the same (e.g., Fig. 1, 2 and 4) while all other figures are also very similar to your JGR 2015 paper. In my opinion such large amount of overlap is surely not warranted in scientific publications.

(1) ANSWER > The submitted paper illustrates numerical simulations carried out by using channels with six different cross sections (Fig. 3). The cross section did not vary in our older paper where we used only one cross section. Running numerical simulations with six different chutes is a significant amount of new and original work in term of design and time. For this reason our new study is a significant and important addition to the scientific literature. The novel and important results of our paper that the reader (and the reviewers) should focus on are summarized by Figs 9, 10 and 12. These figures justify the publication of the paper. The other figures have the purpose to illustrate the system we are studying and, for this reason, they are necessary because, otherwise, the reader would not understand which system our results apply to.

> Here repetitions are of two types: a) concerning the method and b) concerning some of the conclusions. Both are unavoidable and their removal would significantly damage the paper by making it poorer and more difficult to understand. In scientific paper clarity is of paramount importance. The reader does not have time to go back and browse a previous publication where information that is still of key importance here was first introduced. These two types of repetitions are unavoidable for the following reasons.

> Methodological Repetitions. These repetitions are due to the fact that we are using the same materials and geometries of particles and chutes (section 2.1 and Tables 2 and 3 and Figs 1, 2 and 4) and the same numerical modelling (section 2.2) that we have used in our older publication. Both paragraphs and figures have been modified as much as possible where needed. It is however obviously a mistake to remove this information because the reader would not be able to characterize the system we are investigating. This applies also to the other figures in the new paper that show images

of the actual granular flows in motion: they are needed to show which type of granular flows we are talking about.

> Repetitions of Some of the Conclusions. The key concept to understand here is that the effects of channel width, grain size and flow volume are the two (well ... three) sides of the same coin because they interfere with one another. In other words, they must be studied together as demonstrated by the fact that they occur within the same scaling parameter. For example we can ask ourselves what happens to the grain size effect in channels with different widths. Does it change? The same applies to the volume effect. Here we show that these effects are still valid no matter the channel width. For this reason it is not possible to delete the discussion of the grain size effect and the volume effect from the new paper. Importantly, this discussion has also been enriched by our answers to the good comments by reviewer Ming Zhang. The improved version of the paper has been attached to the answers we have provided for Ming Zhang.

> As a general comment we can say that confirmation of previous results is a rare occurrence nowadays but it is of fundamental importance for a healthy science as demonstrated by an inquiry conducted by The Economist where they show that it has been impossible to reproduce most of the scientific findings published recently by academic journals. This is the problem that scientific journals should start tackling very seriously. Everybody who is interested in the future of science should read the article (Unreliable research: Trouble at the lab, Oct 19th 2013, The Economist).

> Concerning the similarity report, we do not know whether this has been obtained by blindly running a computer software, we do however know that the only way to evaluate a paper is by having it read by a human (possibly an informed one).

(2) COMMENT Additionally, it seems that in your new scaling parameter you only replace the length scale in the denominator from width*width to width*length. Is this correct? If so, I would acknowledge that within the paper and tune down the formulation. I would say it is a small update of a scaling parameter.

(2) ANSWER > A change of a quantity in a scaling parameter is not a trivial matter because it affects whether or not it is possible to use this parameter to solve practical problems. Here we show that the channel width that occurs at the denominator of the scaling parameter has an exponent equal to 1 instead of 2. This means that another quantity with the dimension of a length has to be introduced at the denominator of this parameter for it to be dimensionless. We proved here that this quantity is either the length of the flow or that of the deposit.

> We do explain the history of the parameter in section 5.3. This change in the scaling parameter is the result of a long study (more than one year of computer processing time) of the effect of the channel width on flow mobility. As we explain in section 5.3, although in our earlier paper we could guess that the channel width occurs at the denominator of the scaling parameter that the reciprocal of mobility is proportional to, a specific set of investigations where the value of the channel width varies systematically is necessary to figure out its exponent. The paper submitted to SE illustrates this systematic set of investigations.

---

## Referee Comment (RC4) · S. Abe (Referee) · 7 Dec 2016

The manuscript presented by Cagnoli and Piersanti describes an interesting application of the DEM method to a granular flow problem. The estimation of the expected runout distance of granular flows is indeed an important problem in a number of contexts, although the authors might overstate that importance slightly ("...at the top of the list of the most hazardous natural phenomena.", page 1, line 26). There are, however, a number of issues in the manuscript which are not quite clear to me:

Specific Issues:

(1) p. 2, l. 29ff : I can only partially agree with the sentence that DEM is ".. able to estimate correctly the relative energy dissipation of the flows without calibration." . This of course only applies if the interactions implemented in the DEM correctly model the interactions happening in a real granular flow and, importantly, the interaction parameters (friction coefficients, contact stiffness . . .) are correct. And getting the parameters right might, at least in some cases, require calibration. In this context, i.e. whether the chosen interactions are correctly modelling the true mechanics of the granular flow, I would have one fundamental question: Does the lack of dissipation by grain fracturing, which might be relevant in the higher stress regimes of large granular flows, have an influence on the results?

(2) p. 3, l. 10ff. It would be good if the assumption that the behavior of the granular flow is independent of its actual size could be validated by some models.

(3) p. 4, l. 19: What is the justification of using an essentially monodisperse grain size distribution? Does this choice influence the results?

(4) p. 4, l. 31/31: The connection between particle shape and friction angle should be clarified. It is well known that there is such a connection (Abe & Mair, GRL 2009, Kim et al., Geosci. J. 2016) but quantifying it is not trivial.

(5) p. 5, l. 9: Using the density mentioned here, is the grain mass calculated from the volume of the actual grain (i.e. the box shape) or the volume of the spheres forming the contained cluster? How much does this decision influence the aggregate density of the granular material given the difference in the outer dimensions between the cluster and the box-shaped grain?

(6) p. 5, l. 18ff. I fully agree with the authors that the clusters of spheres are an efficient way to model the dynamics of non-spherical particles, in particular regarding the friction in granular materials (Abe & Mair, GRL 2009). However, I would be reluctant to call the

grains used here "angular". At least it should be made clear somewhere early in the manuscript (the introduction maybe) that the authors are approximating angular grains by using aggregates and that they are not using a DEM with true polyhedral particles such as in (Nassauer & Kuna, Granular Matter 2013).

(7) p. 5, l. 23ff: The choice of a Hertzian contact law is an interesting one in this context because it assumes a specific contact shape (i.e. a sphere-sphere contact) whereas the authors are using the sphere clusters to approximate angular grains, which would supposedly lead to quite a different contact shape. Is there any indication if this choice of contact law has a significant influence on the presented results? More technically, is it known if the Hertzian contact law is a better approximation here than a computationally cheaper linear contact law (cf. Donze et al., GJI 1994)?

(8) p. 10, l. 7-9: What about uncertainty due to different initial particle arrangements? Could this be quantified by running a set of models with the same parameters but supplying a different random seed to the particle setup described on top of p. 5?

(9) p. 11, l. 28ff: If I'm reading this paragraph correctly, it seems be to strictly correct only if the curvature of the described "change of slope" is constant for a section of the surface at least as long as the traveling granular avalanche.

(10) p. 13, l. 21-25: Wouldn't the grain size distribution also be a factor influencing the behavior of the flow?

(11) Fig. 4: Looking at the smallest type of grains the authors use, i.e. the ones on the right in Fig. 4, and considering that the contact forces on the grains appear to be calculated based on the inscribed sphere clusters, isn't there a risk that these particles will move by rolling, rotating around their long axis, and therefore produce much lower friction angles than real angular grains?

Technical issues:

(1) p. 2, l. 3/4: I'm not sure that I understand why the fact that some quantities vary

significantly (line 4) automatically causes them to influence the mobility of the flow (line 3).

(2) p. 2, l. 8. It would be good if "mobility" would be defined before being used here.

(3) p. 7, l. 10ff: What is the internal dynamics of the sphere clusters? Are the spheres "glued" together by elastic interactions as in (Abe & Mair, GRL 2009) or are they rigid bodies like the "clumps" in (Cho et al., IJRMMS 2007)?